# OpenReview forum: "A Dual-Modal Framework Utilizing Visual Prompts for Enhanced Patch Analysis"
_ICLR.cc/2025/Conference — Submitted to ICLR 2025_

### Official Review · Reviewer_Myjv · 2024-10-27

**Soundness:** 1
**Presentation:** 1
**Contribution:** 1
**Rating:** 1
**Confidence:** 1

**Summary:**

This paper seems to be nonsensical and generated by LLM.

**Strengths:**

N/A

**Weaknesses:**

N/A

**Questions:**

N/A

**Details Of Ethics Concerns:**

This paper seems to be purely generated by LLM.

---

### Official Review · Reviewer_qkkP · 2024-11-02

**Soundness:** 1
**Presentation:** 1
**Contribution:** 1
**Rating:** 1
**Confidence:** 3

**Summary:**

The authors propose the Image-Guided Code Patch Framework (IGCP) to enhance patch analysis.

**Strengths:**

Patch analysis is an important topic to study and a field where many ML techniques could potentially be explored. The authors also attempt to propose their method from a very theoretical point of view.

**Weaknesses:**

I highly suspect that the whole paper is generated with some LLMs. See the following comments.

1. The presentation of the methodology is unclear and difficult to follow. After careful reading, the proposed methodology remains ambiguous and lacks coherence. The entirety of Section 3 feels like a collection of complex mathematical definitions that seem disconnected from one another, without a clear explanation of how they contribute to the overall methodology.
2. The theoretical analysis is very weak and sometimes lacks clear definition. For example, in theorem 3.1 the final claim is that "If $k$ is characteristic, then the mapping $Z \to \mu_Z$ is injective. However, commonly this claim itself serves as the definition of a kernel $k$ being characteristic. Similar situations happen at multiple other places, where the assumption is too strong to reduce the claim to some very obvious facts.
3. The paper is not clearly finished at various places. For example, in line 351, 355, 365, 368, the equations or texts after the colon is missing. This makes it very hard to comprehend what the authors wanted to express.
4. The experiment setting is very unclear, and I don't understand why all the experiment sections are italicized, making it very difficult to read. In line 428, the author states 'to evaluate the performance of IPPMF'. However, there is no mentioning of what 'IPPMF' is anywhere in the paper.

**Questions:**

See weakness.

**Details Of Ethics Concerns:**

No concern.

---

### Official Review · Reviewer_2PmY · 2024-11-06

**Soundness:** 2
**Presentation:** 2
**Contribution:** 2
**Rating:** 3
**Confidence:** 2

**Summary:**

The paper introduces the Image-Guided Code Patch Framework (IGCP), a novel approach to patch representation learning in software development. Patch representation learning is essential for tasks like bug detection, vulnerability assessment, and code optimization. Existing methods often specialize in either predictive tasks (e.g., security patch classification) or generative tasks (e.g., automated patch description generation), leading to a trade-off between understanding and generation capabilities.

IGCP aims to bridge this gap by integrating code analysis with image processing. The framework treats code patches as probability measures on Polish spaces, providing a rigorous measure-theoretic foundation. This allows for a nuanced understanding of the probabilistic structures inherent in diverse coding domains.

**Strengths:**

I have limited experience in this specific domain, particularly with patch analysis in the context of code. As a result, I find it challenging to fully assess the contributions and strengths of this paper.

**Weaknesses:**

In my view, the paper has some formatting issues, as all text following Section 3.5 appears to be in italics. Additionally, the description of the experimental setup is somewhat lacking, particularly in terms of metric explanations, which are difficult to follow.

**Questions:**

What is the main purpose of Figure 1, and how does it connect to the core content of the paper?

---

### Official Review · Reviewer_BX4B · 2024-11-09

**Soundness:** 1
**Presentation:** 1
**Contribution:** 1
**Rating:** 1
**Confidence:** 3

**Summary:**

This paper first presents Image-Guided Code Patch Framework (IGCP) for patch representation learning for code analysis. Then, the authors introduce several mathematical formulations for IGCP.

The problem is poorly introduced, the technical approach does not seem to be related and the writing looks very unnatural. While it mentions "image-guided", I do not see where image comes from and how it is related to software patches, which is also not well-defined. There are no clear connections between the problem with mathematical frameworks introduced, and how metrics such as BLEU, ROUGE come from. I think this submission is poorly written.

**Strengths:**

N/A.

**Weaknesses:**

See above.

**Questions:**

N/A

---

### Meta-Review · Area_Chair_6hqf · 2024-12-19

**Metareview:**

The paper starts with talking about patch representation learning but rest of the paper is unclear, filled with unrelated concepts written in difficult to understand language. Overall, this paper is not at the level of consideration for ICLR conference and highly likely been generated by an LLM. Therefore, I recommend rejection.

**Additional Comments On Reviewer Discussion:**

All reviewers echoed same concerns as mine and found it difficult to even understand the content of the paper. There was no author rebuttal.

---

### Decision · Program_Chairs · 2025-01-22

Reject